# Examination of *Staphylococcus aureus* Prophages Circulating in Egypt

**DOI:** 10.3390/v13020337

**Published:** 2021-02-22

**Authors:** Adriana Ene, Taylor Miller-Ensminger, Carine R. Mores, Silvia Giannattasio-Ferraz, Alan J. Wolfe, Alaa Abouelfetouh, Catherine Putonti

**Affiliations:** 1Bioinformatics Program, Loyola University Chicago, Chicago, IL 60660, USA; aene@luc.edu (A.E.); tmillerensminger@luc.edu (T.M.-E.); 2Department of Microbiology and Immunology, Stritch School of Medicine, Loyola University Chicago, Maywood, IL 60153, USA; carinermores@gmail.com (C.R.M.); awolfe@luc.edu (A.J.W.); 3Departmento de Microbiologia, Instituto de Ciências Biológicas, Universidade Federal de Minas Gerais, Belo Horizonte 31270-901, Brazil; silvia.giannattasio@gmail.com; 4Department of Microbiology and Immunology, Faculty of Pharmacy, Alexandria University, Alexandria 25435, Egypt; alaa.abouelfetouh@pharmacy.alexu.edu.eg; 5Department of Microbiology and Immunology, Faculty of Pharmacy, Alalamein International University, Alalamein 51718, Egypt; 6Department of Biology, Loyola University Chicago, Chicago, IL 60660, USA

**Keywords:** prophages, *Staphylococcus aureus*, PVL, Egypt

## Abstract

*Staphylococcus aureus* infections are of growing concern given the increased incidence of antibiotic resistant strains. Egypt, like several other countries, has seen alarming increases in methicillin-resistant *S. aureus* (MRSA) infections. This species can rapidly acquire genes associated with resistance, as well as virulence factors, through mobile genetic elements, including phages. Recently, we sequenced 56 *S. aureus* genomes from Alexandria Main University Hospital in Alexandria, Egypt, complementing 17 *S. aureus* genomes publicly available from other sites in Egypt. In the current study, we found that the majority (73.6%) of these strains contain intact prophages, including Biseptimaviruses, Phietaviruses, and Triaviruses. Further investigation of these prophages revealed evidence of horizontal exchange of the integrase for two of the prophages. These Egyptian *S. aureus* prophages are predicted to encode numerous virulence factors, including genes associated with immune evasion and toxins, including the Panton–Valentine leukocidin (PVL)-associated genes *lukF*-PV/*lukS*-PV. Thus, prophages are likely to be a major contributor to the virulence of *S. aureus* strains in circulation in Egypt.

## 1. Introduction

*Staphylococcus aureus* is found in the environment and on the skin and mucus membranes of healthy individuals as a commensal bacterium [1]. However, these bacteria have the potential to cause many forms of infection ranging from mild skin to serious life-threatening infections—including septicemia, pneumonia, and endocarditis [2]. In several countries throughout the world, antibiotic use in not regulated, which leads to increased antibiotic resistance levels [3]. Community-acquired and hospital-acquired *S. aureus* infections are common, and treatment is a major challenge [4,5]. *S. aureus* is able to rapidly acquire antibiotic resistance leading to multi-drug resistant strains such as MRSA (methicillin-resistant *S. aureus*). Of paramount concern is the increase of antibiotic resistance and virulence factor acquisition through mobile genetic elements (MGEs), such as plasmids, transposons, insertion sequences, and phages. Prior studies have identified numerous *S. aureus* virulence factors encoded by MGEs (see review [6]).

Temperate phages that infect *S. aureus* have been intensely studied given their frequent association with virulence [7]. Furthermore, associations between phage activity and pathogenicity have been observed [8]. Numerous *S. aureus* prophages have been identified to date and analysis of their sequences has found genes encoding for Panton–Valentine leukocidin (PVL), exfoliative toxin A (*eta*), and the immune evasion cluster (IEC), which includes the enterotoxin S (*sea*), staphylokinase (*sak*), the chemotaxis inhibitory protein (*chp*), and the staphylococcal complement inhibitor (*scn*) [9,10,11,12,13,14]. This is in addition to several other virulence factors and genes conferring antibiotic resistance. Prior comparative studies of *S. aureus* prophages have found that these phage genomes are highly mosaic [15]. Temperate *S. aureus* phages are generally grouped into one three serogroups (A, B, and F) and one of 12 ‘types’ based upon their integrase gene sequence [16,17,18]. Association between virulence factors, as well as localization of infection, and integrase types have been previously noted [9,15,19,20,21,22,23,24].

In Egypt, antibiotic use is not regulated, and most antimicrobial agents are available without the need for a prescription. Thus, antibiotic therapy is often ineffective, a problem compounded by the use of the wrong antibiotic and both inappropriate dosage and duration. Consequently, Egypt has seen alarming increases in antibiotic resistance, including MRSA and MRSH (methicillin-resistant *S. haemolyticus*) prevalence [25,26,27,28]. Recently, we conducted a genomic study of *S. aureus* isolates from Egypt, contributing 56 new genome sequences to public data repositories [29]. Here, we examined these *S. aureus* genome sequences, as well as 17 publicly available genomes of *S. aureus* isolates also from Egypt. Prophage sequences identified include three different phage genera of the family *Siphoviridae*; more than half encode for one or more virulence factor. Exploring these prophage sequences provides a better understanding of the reservoir of virulence- and antibiotic resistance-associated genes in circulation within Egypt.

## 2. Materials and Methods

### 2.1. Prophage Prediction and Identification

Draft genome sequences isolated from Egypt were retrieved from NCBI. Appendix A lists the accession numbers for these sequences. Each was uploaded to the webtool PHASTER for prophage prediction [30]. PHASTER predictions include incomplete, questionable, and intact prophage sequences. Our analyses focused on intact prophage sequences only. Each intact prophage nucleotide sequence was queried against the viral nr/nt database (viruses (taxid:10239)) via BLAST, and results were recorded.

### 2.2. Cluster Analysis

Homologous prophage sequences were identified using usearch v.11.0.667 [31]. A 50% nucleotide similarity threshold was used to perform clustering using the ‘cluster_fast’ method. Identified clusters were then aligned using the progressiveMauve algorithm [32] and MAFFT v7.388 [33] through Geneious Prime v2019.1.1 (Biomatters Ltd., Auckland, New Zealand). Phylogenetic trees were derived using FastTree v2.1.11 [34] through Geneious Prime and visualized using iTOL v5 [35]. JSpeciesWS v3.4.8 [36] was used to calculate ANI values between prophage sequences.

### 2.3. Pangenome Analysis

Intact prophage sequences were examined using anvi’o v6.2 [37]. A pangenome was computed using the command anvi-pan-genome with the ncbi flag and an mcl inflation of 10. Single copy genes were identified, resulting in a set of 43,152 genes from 496 gene clusters. Pangenome images were generated using anvi’o. Next, results were parsed with Python to create a wedge weighted edge list. This file was input into Cystoscope v3.8.1 (https://cytoscape.org/ (accessed on 1 January 2021)) for visualization.

### 2.4. Gene Annotation

To complement our anvi’o analysis, prophage sequences were also annotated using RAST [38] and examined for antibiotic resistance genes using ResFinder [39]. Furthermore, virulence factors were identified using VFanalyzer [40].

### 2.5. Phylogenetic Analyses

Amino acid sequences for integrase and large subunit terminase were identified in each predicted prophage sequence as follows. Representatives of the Sa1int–Sa2int proteins were retrieved from NCBI and predicted prophage sequences were locally blasted against these sequences using BLAST+. Based on the BLAST results, the protein coding sequence was extracted from the RAST annotation file. These representative integrase type sequences include: NP_510895.1 [Sa1int], NP_058467.1 [Sa2int], NP_803356.1 [Sa3int], YP_002332364.1 [Sa4int], YP_240491.1 [Sa5int], AAX91804.1 [Sa6int], YP_239679.1 [Sa7int], YP_002332477.1 [Sa8int], AAX91428.1 [Sa9int], AAX91273.1 [Sa10int], YP_240184.1 [Sa11int], and YP_001604091.1 [Sa12int]. These representative sequences were selected based upon the strain classification of *S. aureus* prophages previously published by Goerke et al. (2009) [19]. Terminase genes were identified by the RAST annotation. Gene sequences were aligned using MAFFT v7.388 [33] and the phylogenetic tree was derived using FastTree v2.1.11 [34]. Trees were visualized using iTOL v5 [35].

## 3. Results

Seventy-three complete or draft *S. aureus* genomes from isolates collected in Egypt were retrieved from NCBI. These include isolates from blood, aspirate, urine, pus, and sputum [29]. Prophages are abundant within these Egyptian *S. aureus* strains. Fifty-three of the 73 strains investigated harbored recognizable, intact prophages; all of the genomes encoded for phage genes, suggestive of defective or defunct prophages (Appendix A). However, we focused on the 87 intact prophage sequences (Table 1). These intact prophages ranged in size from 13kbp in strain *S. aureus* AA57 to 81kbp in strain *S. aureus* AA70. The average intact prophage length was 41kbp.

Each predicted intact prophage sequence was then queried against the NCBI nr/nt database. All of the predicted prophage sequences exhibited sequence homology to tailed phages of the family *Siphoviridae* (Appendix A). These siphoviruses include three genera— *Biseptimavirus*, *Phietavirus*, and *Triavirus*. While 13 Egyptian *S. aureus* prophage sequences were nearly identical (>90% query coverage and sequence identity) to previously characterized phages, several were distinct. Seven of the predicted prophage sequences shared less than 50% sequence similarity (query coverage) with a characterized phage sequence. These seven include phage_9 (carried by *S. aureus* AA51), phage_19 (*S. aureus* AA4), phage_37 (*S. aureus* AA87), phage_59 (*S. aureus* AA103), phage_61 (*S. aureus* AA78), phage_74 (*S. aureus* 43), and phage_76 (*S. aureus* 23). Phage_19 exhibited greatest sequence similarity (39% query coverage and 96.14% sequence identity) to the virulent phage SA97 [41]. The other six new phages most closely resembled *Staphylococcus* phage SAP090B (Appendix A), which has yet to be isolated or characterized [42].

Our BLAST queries and pangenome analysis suggest that several of the Egyptian *S. aureus* isolates harbor similar prophage sequences. We thus clustered the intact prophage sequences based upon nucleotide sequence similarity, finding eight distinct prophage clusters (Appendix A). Phylogenetic trees were derived for each cluster, such as prophage cluster A (Appendix A). Table 2 summarizes these prophage clusters. While prophage clusters A and D (Biseptimaviruses) and prophage clusters B, C, G, and H (Phietaviruses) have modules of sequence similarity, differences in lengths, gene acquisition, and reassortment events lead to their assignment to separate clusters. Four prophages did not resemble any of the other prophage sequences: phage_8 (*S. aureus* AA93), phage_27 (*S. aureus* AA95), phage_29 (*S. aureus* AA35), and phage_53 (*S. aureus* AA53). We refer to these four prophages as ‘singletons’. Based upon their BLAST queries, we can assign phage_8, phage_27, and phage_29 to Phietaviruses and phage_53 to Biseptimaviruses (Appendix A).

The 87 Egyptian *S. aureus* prophages were next examined for their genic content. While no gene was common amongst all of the prophage sequences, every prophage included at least one gene sequence found within another prophage (Figure 1). The most common gene amongst these prophage sequences, found in 77 of the prophages, is the transcriptional activator *rinB*, which is required for expression of the prophage integrase [43]. As reflected in Figure 2 and Appendix A, the prophages vary in the number of genes encoded (minimum = 17; maximum = 94). Phage_38, from *S. aureus* AA45, contains the most (*n* = 23) unique or ‘singleton’ phage genes.

Next, the prophage sequences were examined for antibiotic resistance genes and virulence factors. Only three prophages were found to encode for an antibiotic resistance gene: phage_35 (cluster E), phage_64 (cluster F), and phage_77 (cluster E) from *S. aureus* AA80, AA70, and 14, respectively; they harbor the tetracycline resistance gene *tet(M)*. Forty-four of the 87 prophages encode for a virulence factor. While the individual virulence factors are listed for each individual strain in Appendix A, the results are summarized in Table 3. The most frequently identified virulence factors were Staphylokinase and SCIN. The majority of the prophages in *Biseptimavirus* prophage clusters A and D encoded both of the related genes *sak* and *scn*, but one member of prophage cluster E (phage_4) and one member of prophage cluster B (phage_7) encode for both genes. Enterotoxin A (*sea*) also was frequently observed, but only within the prophages of clusters A and D. Furthermore, 11 prophages encode for the Panton–Valentine leukocidin (PVL)-associated genes *lukF*-PV/*lukS*-PV (Appendix A). These prophages belong to the genus *Triavirus*: prophage clusters E (*n* = 9) and F (*n* = 2).

Virulence factors have previously been associated with *S. aureus* phage integrase groups. Thus, we identified the integrase coding regions (if present) in each of the 87 Egyptian *S. aureus* prophages. In total, 39 of the 87 prophages were found to include an integrase. A phylogenetic tree was derived (Figure 3). These prophages include integrase type Sa1int, Sa2int, Sa3int, and Sa7int. Sa3int prophages encode for *sak* and *scn* virulence factor-associated genes, as well as several others. One Sa1int prophage, phage_53, also encodes for *sak* and *scn*. PVL-associated genes *lukF*-PV and *lukS*-PV are encoded by some, but not all, of the Sa2int prophages.

To further investigate the relatedness of these prophages, we compared the large subunit of the terminase across all of the prophage sequences (Figure 4). Prophages from the two Biseptimaviruses clusters, A and D, show similarities based upon this protein sequence. Similarities are also shared between Triaviruses and Phietaviruses.

## 4. Discussion

While most of the predicted intact prophages resembled previously characterized staphylococcal phages belonging to the genera Biseptimaviruses, Phietaviruses, and Triaviruses, 7 exhibited less than 50% sequence similarity to sequenced phage sequences indicative of novel gene acquisition. All of the prophages encode for at least one gene shared by another Egyptian *S. aureus* prophage (Figure 2). Genes essential to lysogeny of the bacterial host were frequently identified within the predicted prophage sequences, including integrases and *rinB*, which is found in the vast majority of *S. aureus* siphoviruses [44]. Nevertheless, our pangenome analysis uncovered several isolates encoding for genes unique among the Egyptian *S. aureus* prophages (Figure 1). Prior research looking at Staphylococcal phages found that horizontal gene transfer amongst these phages is frequent [45].

As expected, prophages belonging to the same cluster have more genes in common (Figure 1). Overall, the integrase groups align with the taxonomic family. Sa3int phages belong to prophage clusters A and D, members of *Biseptimavirus*. Sa1int phages belong to *Phietavirus* prophage clusters B and C and the *Phietavirus* singleton phage_29. Sa1int also includes the singleton phage_53, which most closely resembled a *Biseptimavirus* (61% query coverage, 99.98% sequence identity). The Sa2int group includes 15 Triaviruses (prophage clusters E and F) and 1 *Phietavirus*, phage_19 from prophage cluster G. While the BLAST analysis confirms the taxonomic grouping of phage_53 and phage_19 (Appendix A), the integrase gene analysis suggests that they exchanged integrase genes with a *Phietavirus* and *Triavirus*, respectively, over their evolutionary history. Prior research found that temperate phage within the same Int group are more likely to exchange genetic modules with each other than with phage outside of their group [9]. The Egyptian *S. aureus* prophages concur with this finding; genes are more commonly shared between prophages belonging to the same Int group (Figure 2).

Prophage cluster H includes helper-phage sequences. This distinction is made based upon their BLAST sequence homology to the well-studied *S. aureus* phages φ11 and φ80α. The prophage sequences of cluster H are found in *S. aureus* strains AA30, AA32, AA45, AA68, and AA77. Helper-phage, similar to φ11 and φ80α, have been observed within *S. aureus* isolates, playing a key role in bacterial pathogenicity [7,46]. These helper-phages interact with *S. aureus* pathogenicity islands (SaPIs), aiding in excision and replication after a helper-phage is induced or a superinfection of helper-phage takes place [7]. Evidence suggests that helper-phage and SaIPs coevolve, losing and gaining resistance rapidly [47].

The Egyptian *S. aureus* prophages carry few antibiotic resistance genes. Prior studies of *S. aureus* genomes have similarly found that phages rarely carry antibiotic resistance genes (see review [48]). The Egyptian *S. aureus* prophages do, however, encode for several different virulence factor genes—including serine proteases, staphylokinases, chemotaxis inhibitory proteins (CHIPS), Staphylococcal complement inhibitors (SCIN), and toxins. The pattern of virulence factor presence/absence observed is indicative of acquisition via horizontal gene transfer and/or recombination, as has previously been noted [49]. Most frequently detected are the staphylokinase *sak* and SCIN *scn* genes, part of the IEC, and enterotoxin S *sea* (also part of the IEC). Surveys of MRSA strains in Saudi Arabia and Libya similarly found high prevalence of *sak* and *scn* among strains [50,51]. Previous studies have found that the IEC is frequently found within Sa3int prophages [19], and all of the Sa3int type prophages identified amongst the Egyptian isolates encode for *sak* and *scn* (Figure 3). While the integrase sequences of the Sa3int Egyptian prophages are nearly identical (>99.5% identical), half contain *sea*, *selk*, and *selq*, while the other half do not. The distinction between these two ‘groups’ does not correspond with their cluster, as members of both prophage clusters A and D encode for *sea*, *selk*, and *selq*. Furthermore, the presence/absence does not correspond with the bacterial isolation site nor if the *S. aureus* strain is hospital- or community-acquired [29]. Rather, this distinction corresponds with the phylogenomic tree for the Egyptian *S. aureus* host [29].

The most well studied phage-mediated virulence factor for *S. aureus* is PVL, which targets human phagocytes [10,13]. PVL is a bacteriophage-encoded bi-component pore-forming leukotoxin, *lukS*-PV and *lukF*-PV, which is secreted and assembled into a ring shape complex that acts as a membrane pore [52]. PVL is typically carried by community-associated MRSA strains [53,54,55]. Nine of the 11 PVL positive strains are *mecA* positive; *S. aureus* AA68 and AA69, which harbor phage_16 and phage_11, respectively, do not encode for *mecA*. Furthermore, 6 of the prophages encoding *lukS*-PV and *lukF*-PV have the SaInt2 type integrase (Figure 2). This concurs with that prior research finding that *lukS*-PV and *lukF*-PV are typically carried by SaInt2 type prophages [19]. Integrase genes could not be identified in the other five prophage sequences. It is worth noting, however, that only 13% of the intact prophages carried *lukS*-PV and *lukF*-PV. Similar studies of temporally and geographically related *S. aureus* strains have found a low occurrence of PVL positive strains [51,56].

## 5. Conclusions

Our examination of *S. aureus* strains within a single hospital and within a single region reveals a diverse group of prophages in circulation. While these bacteria harbor prophages belonging to three different genera of *Siphoviridae*, comparison of the prophage sequences revealed numerous likely events of horizontal gene transfer. Prophages are abundant among the Egyptian isolates and over half of these prophages encode for virulence factors, including PVL. This is particularly concerning given the rise of antibiotic resistant pathogens, including *S. aureus* and MRSA strains, in Egypt and the region. We conclude that prophages are likely to be a major contributor to the virulence of strains in circulation.

## Figures and Tables

**Figure 1 viruses-13-00337-f001:**
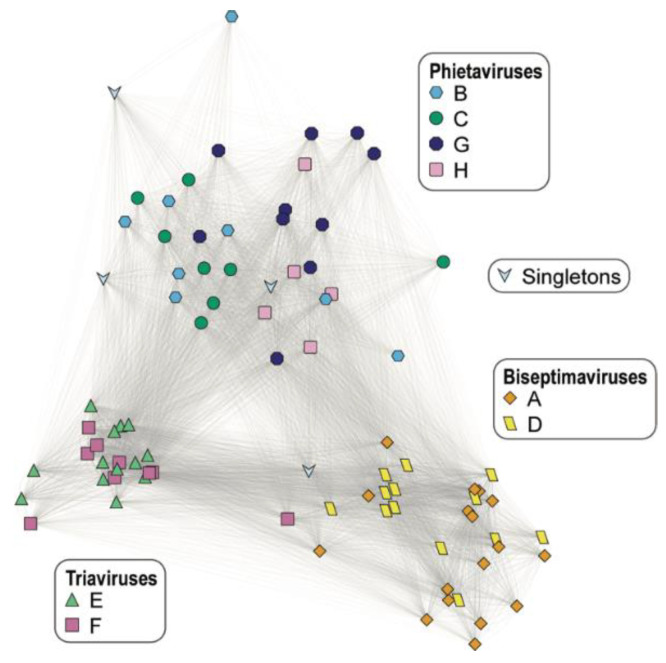
Prophage network of shared genes. Each node corresponds with a single predicted prophage sequence. The shape and color of the node represents the identified prophage cluster for the sequence. Two nodes are connected by an edge if they both share a common gene. The weight of the edge represents the number of common genes between two prophages.

**Figure 2 viruses-13-00337-f002:**
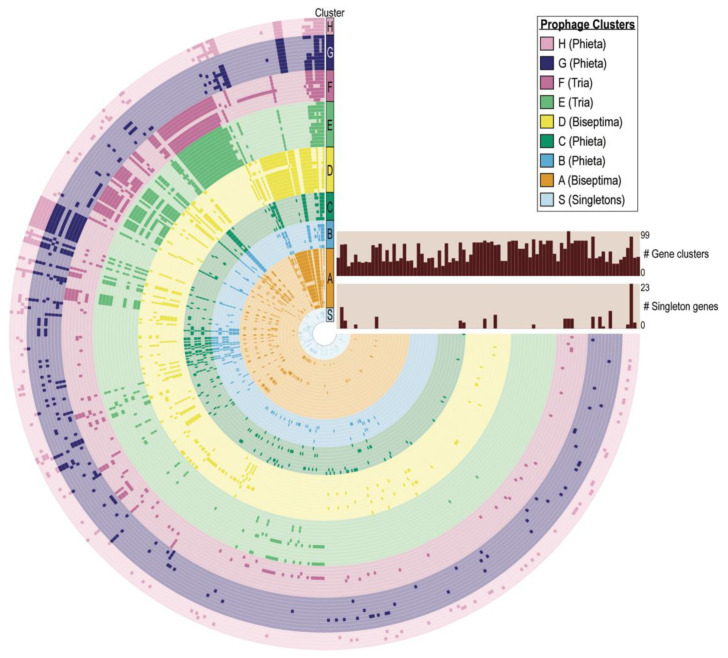
Pangenome of Egyptian *S. aureus* prophages. Each ring in the graph represents an individual *S. aureus* prophage sequence, color coded according to the assigned prophage cluster. Each ray in the graph indicates the presence (darker coloration) or absence (lighter coloration) of a given homolog. The number of gene clusters (no. of CDS) and singleton genes (unique genes, i.e., no homologs within other prophage sequences) found within each prophage sequence are shown in the two bar charts.

**Figure 3 viruses-13-00337-f003:**
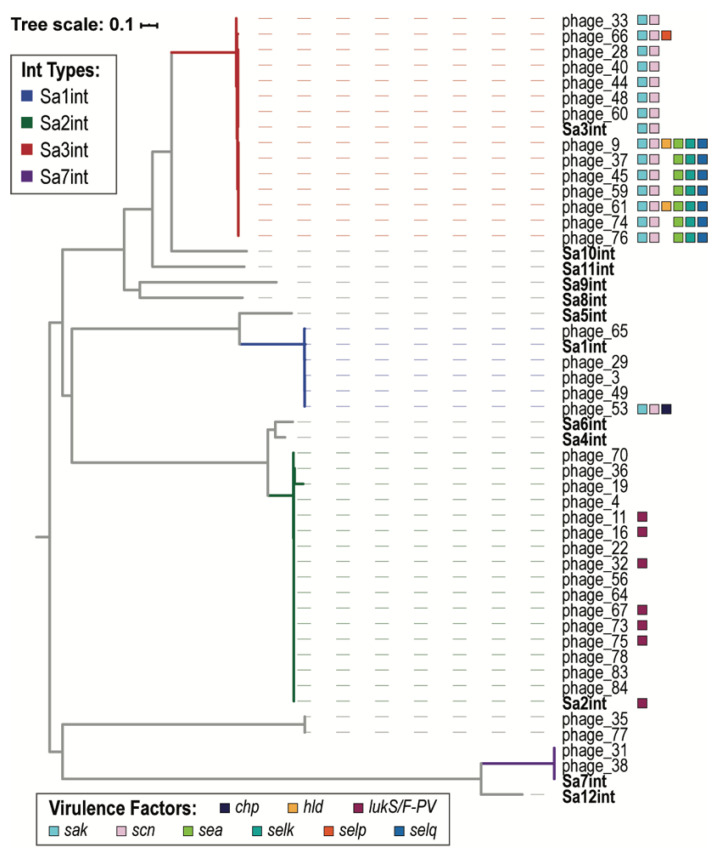
Integrase phylogenetic tree. Representative sequences of the 12 *S. aureus* integrase types (Sa1int–Sa12int) are also included in the tree, shown in bold. Sa1int, Sa2int, Sa3int, and Sa7int branches are colored blue, green, red, and purple, respectively. Virulence factors are indicated for the Egyptian *S. aureus* prophage sequences and the Sa1int, Sa2int, Sa3int, and Sa7int reference sequences.

**Figure 4 viruses-13-00337-f004:**
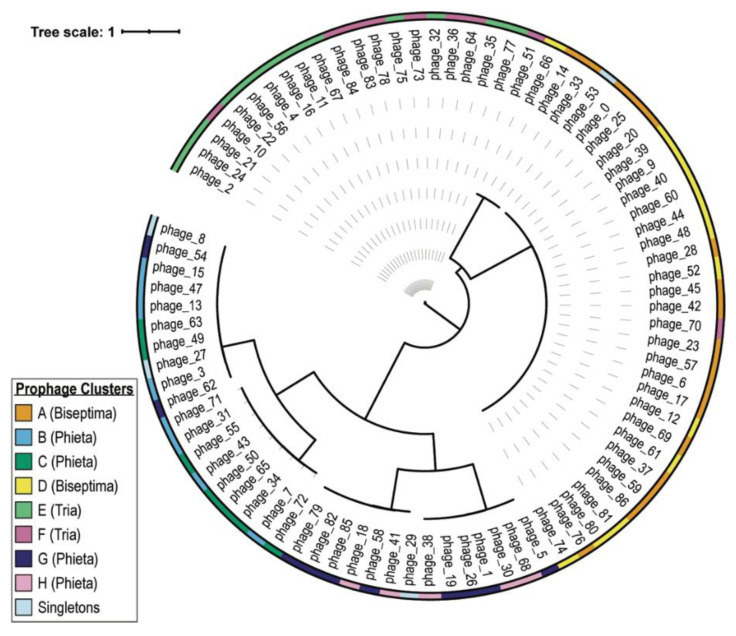
Phylogenetic tree of the terminase large subunit amino acid sequences. Prophage clusters are indicated as are the predicted genera (Biseptimaviruses, Phietaviruses, and Triaviruses).

**Table 1 viruses-13-00337-t001:** Summary statistics of Egyptian *S. aureus* genomes and their prophages.

Study	No. Strains with Intact Prophages	No. Intact Prophages	Average No. Prophages/Strain	Max No. Prophages/Strain
Medical Microbiology Laboratory at AMUH isolates (*n* = 56)	45	71	1.5	3
Other MRSA isolates from Egypt (*n* = 17)	8	16	2	3

**Table 2 viruses-13-00337-t002:** Prophage clusters amongst Egyptian *S. aureus* isolates.

Cluster ID	Cluster Size	ANI Score Range (%)	Predicted Genus
A	17	87.93–100	*Biseptimavirus*
B	8	73.29–100	*Phietavirus*
C	8	82.92–99.86	*Phietavirus*
D	13	91.12–100	*Biseptimavirus*
E	13	96.09–100	*Triavirus*
F	9	88.39–100 *^a^*	*Triavirus*
G	10	77.01–100	*Phietavirus*
H	5	92.72–97.19	*Phietavirus*

*^a^* Sequence divergence between phage_10 and phage_70 exceed the threshold for ANI calculations and their pairwise comparison is not included in the reported range.

**Table 3 viruses-13-00337-t003:** Virulence factors encoded by Egyptian *S. aureus* prophages.

VF Class	Virulence Factors	Related Genes	No. Prophages
Enzyme	Serine protease	*splA*	2
		*splB*	2
		*splC*	2
		*splD*	1
		*splE*	1
		*splF*	1
	Staphylokinase	*sak*	30
Immune Evasion	CHIPS	*chp*	1
	SCIN	*scn*	30
Toxin	Delta hemolysin	*hld*	2
	Enterotoxin A	*sea*	17
	Enterotoxin G	*seg*	1
	Enterotoxin I	*sei*	1
	Enterotoxin Yent1	*yent1*	1
	Enterotoxin Yent2	*yent2*	1
	Enterotoxin-like K	*selk*	7
	Enterotoxin-like M	*selm*	1
	Enterotoxin-like N	*seln*	1
	Enterotoxin-like O	*selo*	1
	Enterotoxin-like P	*selp*	1
	Enterotoxin-like Q	*selq*	7
	Gamma hemolysin	*hlgA*	3
	Leukotoxin D	*lukD*	3
	Leukotoxin E	*lukE*	3
	Panton-Valentine leukocidin	*lukF*-PV	11
		*lukS*-PV	11

## Data Availability

All data used in this study is publicly available. Accession numbers are listed in Appendix A.

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
