# Peer review of "Examination of Staphylococcus aureus Prophages Circulating in Egypt"

_viruses, 2021, doi:10.3390/v13020337_

Round 1

Reviewer 1 Report

This manuscript describes the genomic analysis of more than hundred staphylococcal prophages found in S. aureus genomes identified in Egyptin hospital. Most of this phages, with the exception of 7, exhibit high identity (>90%) with already described genomes. The paper is well written and illustrated. The analysis is pertinent. 

It would have been interesting to check the 7 new phages for possible lytic behaviour when activated. What is the situation for those close to known phages?

Minor detail: indicate in the introduction that these phages are Siphoviridae.

Author Response

Thank you for your review.

COMMENT: It would have been interesting to check the 7 new phages for possible lytic behaviour when activated. What is the situation for those close to known phages?

REPLY: Fantastic suggestion. We probed deeper into these 7 phages and the phages that they most closely resemble. 6 of the phages show greatest homology (39-41% query coverage) to Staphylococcus phage SAP090B. This phage was identified in sequence only, i.e. it was not isolated or characterized. As such, these are phages that would be really interesting to characterize. However, at this time, we have limited information. The 7th phage, phage_19, most closely resembles (39% query coverage and 96.14% sequence identity) the virulent phage SA97. We have included a citation to this work. Phage_19, however, differs significantly from this phage (per the query coverage). Further investigation of phage_19 would also be interesting follow up work.

We have included the above details in the manuscript (lines 135-138), including related citations.

COMMENT: Minor detail: indicate in the introduction that these phages are Siphoviridae.

REPLY: We have made this change. (lines 69-70)

Reviewer 2 Report

Manuscript submitted by Ene et al., examined prevalence of Staphylococcus aureus  antibiotic resistance mediated by  Prophages in Egypt. Authors have conducted genomic study of  several S. aureus isolates from Egypt  and thoroughly examined genomic sequences of the isolates. Based on their findings, authors concluded that prophages are key contributor to the virulence of strains in circulation in Egypt. 

I find the manuscript is clearly written in a professional unambiguous language, properly referenced and addressing important problem of antibiotic resistance. I commend the authors for their extensive data set and analysis. 

Author Response

Thank you very much for your kind words and for taking the time to review our manuscript.